# Discovering Candidate Anti-Aging Perturbations Using a Foundation Model for Gene Expression

**DOI:** 10.3390/ijms262411977

**Published:** 2025-12-12

**Authors:** Erik Tadevosyan, Evgeniy Efimov, Dmitrii Kriukov, Ekaterina Khrameeva

**Affiliations:** Center for Biomedical Technologies, The Territory of the Skolkovo Innovation Center, 121205 Moscow, Russia; erik.tadevosian@skol.tech (E.T.); evgeniy.efimov@skol.tech (E.E.); dmitrii.kriukov@skol.tech (D.K.)

**Keywords:** aging, single-cell gene expression, AI tools in bioinformatics

## Abstract

Aging is a progressive functional decline driven by complex genetic, epigenetic, environmental, and stochastic interactions that traditional linear models struggle to capture. Using human single-cell RNA-seq data from the multi-tissue AgeAnno dataset, we fine-tuned scGPT, a large transcriptomic model, to predict chronological age groups, achieving high classification accuracy. To identify genes influencing age predictions, we systematically perturbed individual genes in silico and quantified their effects, classifying them as pro- or anti-aging candidates. Our results demonstrate that scGPT does capture age-related dependencies in single-cell data and can be utilized to discover novel candidate gene perturbations—potential targets to be validated as anti-aging interventions.

## 1. Introduction

Aging is a multifactorial process characterized by progressive functional decline and increasing vulnerability to disease, driven by complex, nonlinear interactions among genes, proteins, metabolites, and environmental factors [1]. To measure aging, researchers have developed many approaches based on diverse biological data types, including gene expression profiles [2]. Previous transcriptome-based models showed that gene expression can effectively predict chronological age, but these models were typically linear and thus unable to capture the nonlinear, high-dimensional nature of gene regulatory networks [3]. Recently, deep learning approaches and large transcriptomic foundation models have emerged, such as transformer-based scGPT [4].

These models are designed to generalize across tissues, species, and experimental protocols by learning latent representations of gene co-expression and context-dependent regulation. When fine-tuned for downstream classification tasks, a small feed-forward neural layer, often referred to as a classification head, is attached on top of the pretrained transformer encoder. This head maps cell-level embeddings produced by the transformer into a probability distribution over discrete target classes, such as tissue identity, disease state, or, in this study, age group. By retaining the pretrained contextual embeddings, while optimizing only the parameters of the classification head, fine-tuning enables the model to leverage large-scale prior biological knowledge, while efficiently adapting to the specific prediction task. This design allows scGPT to capture higher-order regulatory dependencies and subtle shifts in transcriptional programs that accompany aging, potentially providing more biologically grounded insights than conventional linear or shallow architectures.

Despite their promise, scGPT-like models have not yet been applied to core problems in aging research—specifically, identifying transcriptomic patterns of aging and nominating candidate genes for potential interventions aimed at modulating human lifespan. Understanding which genes causally influence the aging process remains a central challenge in geroscience. Traditional correlation-based approaches can identify genes whose expression changes with age but cannot separate causal drivers from downstream effects. In contrast, in silico perturbation analysis allows us to simulate gene up- or downregulation, observe the resulting shift in predicted cellular age, and thus identify genes potentially playing a regulatory role in age-associated transcriptional programs. While all simulated perturbations represent computational predictions and should not be interpreted as evidence of causal biological effects, they help to generate hypotheses and prioritize candidate genes for future experimental validation.

In this study, we fine-tune scGPT on the AgeAnno dataset [5] to evaluate its ability to distinguish between younger and older cells and to explore how simulated changes in the expression of individual genes affect the model’s age predictions. After training, we digitally increase or decrease each gene’s expression and observe the corresponding shift in predicted age, allowing us to identify genes whose expression patterns are closely associated with aging-related transcriptional changes. Although these simulations do not imply biological causality, they provide a practical framework for generating testable hypotheses regarding genes potentially involved in aging-related transcriptional dynamics.

## 2. Results

### 2.1. Fine-Tuning scGPT for Age Classification

We fine-tuned the scGPT model for binary age classification (Figure 1a), distinguishing between cells retrieved from the mid-age (20–59 years) and the old-age (60–100 years) donors, as defined in the AgeAnno dataset. The input gene set for model fine-tuning comprised 869 genes reported as age-associated in the original AgeAnno publication [5].

### 2.2. Model Performance

Across all tissues, the fine-tuned scGPT model yielded generally comparable performance metrics, with consistent ROC-AUC values. While certain tissues (such as stomach and blood) exhibited more pronounced differences in model performance, the aggregate results across tissues showed that scGPT achieved a marginally higher average ROC-AUC of 0.91 compared to 0.89 for logistic regression (LR) applied to the same dataset and gene subset (Figure 1b).

Importantly, this modest improvement does not imply that scGPT is inherently superior—or inferior—to LR for predicting chronological age. Rather, it demonstrates that scGPT achieves performance on par with a standard baseline model, thereby validating its suitability as a tool for downstream analyses. Specifically, this confirms that the fine-tuned scGPT can reliably be used to investigate the individual contribution of each gene to age-related transcriptional changes, including through in silico perturbation experiments.

### 2.3. Perturbation Analysis Identifies Candidates for Pro- and Anti-Aging Gene Perturbations

To investigate which genes contribute most to age predictions, we conducted an in silico perturbation analysis in each tissue. Because extracting gene-level importance from transformer models is non-trivial, we performed the following experiment: for each gene, predictions were obtained from the fine-tuned scGPT model under two conditions: (1) setting the gene’s expression to zero (simulating a knockout) and (2) setting the gene’s expression to its maximal value (simulating strong overexpression) (see Materials and methods for details). This procedure yielded per-tissue lists of predicted perturbation candidates, along with their adjusted *p*-values (Figure 1c; Appendix A). Importantly, these predictions require further validation before they can be considered as robust candidate perturbations.

### 2.4. Predicted Candidates Are Stable Across Repeated Subsampling

To evaluate the robustness of the predicted candidate perturbations, we repeated the perturbation identification procedure under multiple rounds of random subsampling in each tissue (see Materials and Methods). The top-ranked genes showed highly consistent perturbation directions and statistical significance across iterations (Appendix A). Ranking genes by their stability revealed that many remained significant (adjusted *p* < 0.05) in all subsampling runs, and the estimates of their effect exhibited uniformly small standard deviations, indicating strong reproducibility of both direction and magnitude (Figure 1c and Figure A2). These findings support the stability and statistical robustness of the perturbation results. Comparison of the top-ranked genes with the Open Genes database [6] of experimentally supported aging-related genes shows a substantial overlap (33–50% of genes significant in all subsampling runs per tissue, Figure 1c and Figure A2), further supporting the biological relevance of obtained predictions. However, we note that these perturbation outcomes still represent model-based predictions and do not imply biological causation. They therefore serve as hypothesis-generation tools rather than mechanistic claims.

Thus, we present a methodological framework for identifying candidates for pro- and anti-aging gene perturbations across tissues, demonstrating the potential of large transcriptomic models such as scGPT to capture age-related patterns and generate biologically meaningful hypotheses in aging research (Table 1).

## 3. Discussion

We present the first example of using a foundation model for gene expression in aging research to identify candidates for gene perturbations—pro- or anti-aging ones. Importantly, this study should not be viewed as intended to produce a robust list of anti-aging candidate genes. Instead, it serves as a methodological proof-of-concept showing that large transcriptomic models like scGPT can be used in principle not only as predictive tools but also as instruments for hypothesis generation in aging research.

The modest performance gap in prediction metrics between scGPT and LR likely reflects that both models are trained on a curated set of aging-associated genes, which already provides a strong age signal, allowing even simple linear models to achieve high accuracy. This pattern is consistent with prior works, where relatively simple models often perform competitively with more complex architectures [7,8,9,10]. Furthermore, both the BiT age [8] and the recent DeepQA framework [11] suggest that performance gains from more complex architectures can be modest when age-informative genes are preselected. In this context, scGPT’s advantage lies not in maximizing AUC but in its ability, learned during pre-training, to implicitly model the gene regulatory network [12], thereby capturing how perturbing a single gene propagates through the transcriptome even when that gene lacks a direct linear association with age. We validate this approach by confirming that scGPT’s representations retain the cellular age signal as effectively as raw transcriptomic data (as measured by LR), ensuring its suitability for identifying potential pro- and anti-aging gene candidates. Accordingly, the value of scGPT in this work is reflected primarily in its hypothesis-generation capabilities rather than in an improvement in classification accuracy.

Beyond methodological perspective, our study illustrates how perturbation analysis can be used to provide insights into the high-order structure of aging-related gene regulation. By simulating the up- or downregulation of individual genes and observing corresponding shifts in predicted age, we can infer which genes are most influential in shaping transcriptomic aging signatures. This analysis is designed to map how local gene expression changes propagate through the latent representation learned by the model, revealing regulatory leverage points that may control broader aging-related programs. However, it is important to emphasize that the perturbation outcomes do not imply biological causality. The observed shifts in predicted age represent model-predicted associations learned from training data, not experimentally validated mechanistic effects. These outputs should therefore be interpreted solely as hypothesis-generating signals, guiding downstream experimental prioritization rather than providing causal conclusions.

The proposed framework also complements existing systems-biology methods. Integrating the predicted genetic perturbations effects with pathway and network analyses could reveal aging-relevant subnetworks and highlight best candidates for downstream experimental validation. Such integration would help connect the predictive capabilities of deep learning models with mechanistic insight derived from empirical biology. In this sense, scGPT acts as a hypothesis-generating engine, narrowing the search space for follow-up validation and accelerating discovery in geroscience.

Nevertheless, several limitations must be acknowledged. First, although scGPT can rank genes by predicted “pro-” or “anti-aging” effects, its black-box nature prevents explaining these choices or revealing underlying mechanisms. Second, scGPT requires substantial computational resources; e.g., our analysis was limited to a subset of 869 genes to fit within reasonable memory constraints. Finally, a conceptual limitation is the usage of chronological age labels provided in the AgeAnno dataset. Chronological age does not account for individual medical history, lifestyle, or individual molecular aging trajectories. Future work should incorporate biologically informed aging labels (e.g., frailty index, intrinsic capacity score, or biological age [13,14,15]) when such data become available for gene expression at single-cell resolution.

Looking forward, future research could explore multi-omics integration to combine transcriptomic perturbations with epigenetic or proteomic data, enhancing mechanistic resolution. Another promising direction is cross-species transfer learning, where foundation models pre-trained on diverse organisms could uncover evolutionary conserved regulators of aging. Furthermore, incorporating explainability methods, such as feature attribution (e.g., saliency mapping, attention and rationale models [16]) could help dissect which biological processes or pathways contribute most to the model’s predictions.

Most importantly, future work should focus on developing rigorous validation strategies for anti-aging perturbations predicted by large models. As the first step, we propose testing predicted genes against existing independent datasets of anti-aging interventions using causal inference approaches that statistically evaluate whether effects persist across datasets [17]. As the second step, top candidates should be confirmed through classical experimental perturbations such as gene knockdowns or overexpression in cell lines and model organisms.

## 4. Materials and Methods

### 4.1. Dataset

We used the AgeAnno dataset [5], comprising scRNA-seq data from human tissues obtained from donors whose chronological ages ranged from 0 to 100+ years. The exact chronological ages are binned into four age groups in the dataset [5]: “youth” (0–19), “mid” (20–59 years), “old” (60–100 years), and “supold” (>100 years), from which we used only the mid and old age groups due to limited numbers of donors in other groups (Figure A1; Table A1). We omitted tissues with low (<3) sample coverage per age group, resulting in 9 tissues (blood, skin, brain, lung, skeletal muscle, stomach, bladder, liver, and bone marrow). Raw count matrices were filtered with the scanpy [18] Python package (version 1.11.5). Genes expressed in fewer than 1% of cells were excluded. Gene expression values were discretized into 51 bins following the preprocessing protocol established by the original scGPT authors [4]. This binning strategy is a core component of the scGPT input representation and was applied as recommended in the official documentation and reference implementation (https://github.com/bowang-lab/scGPT/, accessed on 25 October 2025).

The total number of cells included in the analysis for each tissue and age group is provided in Table A1.

For each tissue type, the training and testing splits were performed at the donor level (i.e., all cells from a given donor were assigned exclusively to either the training or the test set). This donor-wise partitioning prevents data leakage and ensures that the model is evaluated on truly unseen individuals. No within-dataset batch correction was applied. All experiments were conducted with a fixed random seed to guarantee reproducibility across runs.

For the preliminary complexity reduction step, we subsampled the gene set by selecting genes previously reported (in AgeAnno) to be significantly associated with aging in the comparison between mid-age and old-age groups (adjusted *p*-value < 0.05), further filtered to exclude genes exhibiting tissue-specific or cell-type-specific expression changes, and then intersected the filtered subset with the scGPT training genes, resulting in total 869 genes (Appendix A).

### 4.2. scGPT Fine-Tuning Configuration

During fine-tuning, we optimized only the parameters of the classification head, while keeping the pretrained transformer encoder weights entirely frozen. This strategy allowed the model to retain general transcriptomic representations learned during pretraining while adapting specifically to age prediction. All computations were performed on an NVIDIA RTX 4090 GPU (Santa Clara, CA, USA) with 24 GB of memory. We used the original scGPT model without any modifications to its core architecture. Only a custom fully connected prediction head was trained on top of the frozen scGPT encoder. To obtain a single embedding representing each cell, we aggregated the gene-level embeddings using average pooling. The prediction head consisted of five linear layers, each with 1024 hidden units, interleaved with GELU activation functions. Layer normalisation was applied immediately after the scGPT embeddings, and a dropout rate of 0.4 was used throughout the head.

The model was trained for 20 epochs with a batch size of 256 and a learning rate of $1\times 10^{-4}$. We employed the AdamW optimizer with a weight decay of $1\times 10^{-7}$ and used the binary cross-entropy with logits loss (BCEWithLogitsLoss) with class-specific weights to account for label imbalance [19].

The underlying scGPT configuration remained unchanged.

### 4.3. Logistic Regression Configuration

Logistic regression (LR) was employed with default parameters and L1 regularization using the scikit-learn Python package (version 1.7.2) [20]. The model was configured as a pipeline consisting of two components: a feature scaling step (StandardScaler function) and a multinomial logistic regression classifier (LogisticRegression function with parameters: random_state = 42, max_iter = 1000, multi_class = ‘multinomial’).

The inclusion of the StandardScaler ensured that each feature had zero mean and unit variance, improving model convergence and stability. The LR model was trained using the same input features as the scGPT fine-tuning experiment, allowing for a fair comparison of predictive performance.

### 4.4. Identification of Putative Pro-Aging and Anti-Aging Perturbations

For the perturbation analysis, 300 cells were randomly selected per tissue (150 from the mid-age group and 150 from the old-age group) to balance numbers of cells per tissue per age group. Downsampling numbers of cells per tissue is an important step to avoid potential confounding from tissue composition. For every gene of interest, predictions were obtained from the fine-tuned scGPT model under two simulated perturbation conditions: knockout and overexpression. In silico perturbations were performed for all genes included in the model training set, as well as for additional curated aging-related genes from the OpenGenes database [6] (restricted to entries with “high” or “highest” confidence levels) and the GenAge database [21]. This combined gene set enabled a comprehensive assessment of how both model-informed and literature-validated aging-associated genes influence predicted chronological age under knockout and overexpression conditions.

To simulate these perturbations, we applied a gene-specific strategy in which the maximal raw expression value of each gene was identified across all cells in a particular tissue. Perturbed profiles were then generated by replacing the gene expression value with zero to approximate knockout, or with its empirical maximum to approximate strong overexpression. This procedure follows the perturbation logic used in existing frameworks such as CellOracle [22] and ensures that all simulated perturbations remain within the observed range of the training data. These perturbations represent intentionally extreme manipulations and are therefore intended solely for hypothesis generation rather than for quantitative estimates of experimentally observed expression changes.

Model outputs corresponded to the predicted probability distribution across age groups for each cell, allowing us to estimate how each perturbation shifted the predicted cellular age. Subsequently, *p*-values were computed for each gene using Mann–Whitney statistical test, followed by Holm correction. Genes showing a statistically significant difference between age groups, as well as $\frac{P(\mathrm{old}\;\mathrm{class}\;\mathrm{probability}\;\mathrm{with}\;\mathrm{low}\;\mathrm{expression})}{P(\mathrm{old}\;\mathrm{class}\;\mathrm{probability}\;\mathrm{with}\;\mathrm{high}\;\mathrm{expression})}>1$ were classified as candidates for anti-aging perturbations, whereas genes with $\frac{P(\mathrm{old}\;\mathrm{class}\;\mathrm{probability}\;\mathrm{with}\;\mathrm{low}\;\mathrm{expression})}{P(\mathrm{old}\;\mathrm{class}\;\mathrm{probability}\;\mathrm{with}\;\mathrm{high}\;\mathrm{expression})}<1$ were classified as candidates for pro-aging perturbations.

We emphasize that the perturbation procedure reflects changes in the scGPT model’s output and cannot serve as an evidence of biological causality. The results represent model-derived predictions, not experimentally validated effects.

### 4.5. Estimating Stability of Predictions

To evaluate the robustness of our procedure for identifying putative pro-aging and anti-aging perturbations, we performed a stability assessment experiment. In the perturbation identification workflow described above, random subsets of 300 cells per tissue are sampled for analysis. We repeated this procedure 10 times, each time drawing a new independent subset of 300 cells from each tissue, thereby generating 10 lists of pro-aging and anti-aging candidates per tissue. For each gene, stability was quantified as the proportion of iterations (out of 10) in which the gene was classified as “pro-aging” or “anti-aging,” respectively, with an adjusted *p*-value < 0.05. This proportion provides an empirical measure of the robustness of both the inferred perturbation direction and its statistical significance across subsampling iterations.

### 4.6. Comparison with the Open Genes Database

We compared the predicted pro-aging and anti-aging candidates with the Open Genes database [6] which contains manually curated evidence-based confidence levels for genes associated with aging. The database was downloaded in tab-separated format, containing 2405 human genes annotated with confidence levels ranging from “highest” to “lowest” based on published experimental evidence.

## 5. Conclusions

Our findings demonstrate that scGPT can capture aging-related transcriptional patterns and enable systematic in silico perturbation analysis at the gene level. Although its predictive performance is similar to that of simpler models, scGPT’s key advantage lies in its ability to capture latent regulatory structure and generate hypotheses about genes that may underlie aging-associated transcriptomic changes. These predictions should, however, be interpreted with caution, as they remain computational inferences and require careful downstream experimental validation. Future work should expand dataset coverage (including phenotypes that better approximate biological aging than chronological age), integrate higher-confidence annotations, and incorporate causal inference frameworks for evidence aggregation across datasets. Ultimately, definitive assessment will require targeted experimental validation. By outlining a perturbation-based interpretability workflow for a large transcriptomic model, this study illustrates both the potential and current limitations of applying foundation models to uncover new biological insights into aging.

## Figures and Tables

**Figure 1 ijms-26-11977-f001:**
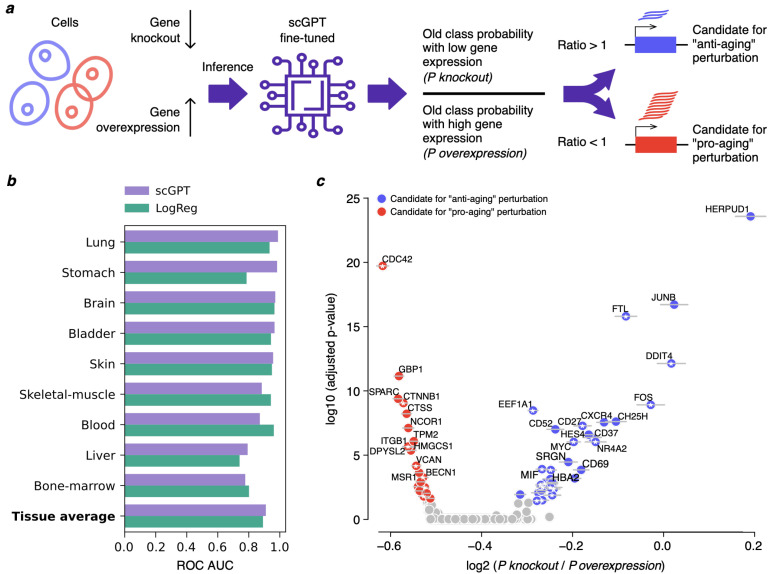
Fine-tuned scGPT model captures age-related transcriptomic patterns in the AgeAnno dataset. (**a**) Schematic of the proposed in silico perturbation analysis for identifying candidates for pro- and anti-aging perturbations. (**b**) ROC AUC values illustrating classification accuracy of scGPT and LR models on aging-associated genes from the AgeAnno dataset (mid-age versus old-age cells). (**c**) Example of model output for blood, showing predicted pro-aging (red) and anti-aging (blue) gene candidates, subject to thorough post-hoc validation. Colored dots highlight genes with significant adjusted *p*-values across 10 out of 10 random subsampling iterations, while gray horizontal bars represent standard deviations across the same iterations. X and Y axes show average log2 ($P_{knockout}/P_{overexpression}$) and maximal log10 (adjusted *p*-value) values across all iterations, respectively. Mitochondrial and heat-shock genes are not shown, as their expression may reflect technical artifacts of scRNA-seq protocol. White stars at dot centers mark known aging-associated genes according to the Open Genes database [6]. Other tissues are shown in Figure A2.

**Table 1 ijms-26-11977-t001:** Comparison of the scGPT-based framework with classical transcriptomic aging models. While classical models offer statistical transparency but limited mechanistic insight, scGPT provides functional interpretability through in silico perturbation, enabling simulation of gene regulation effects and generation of biologically testable hypotheses.

Aspect	Classical Approaches (LR, Elastic Net, etc.)	scGPT (Foundation Model)	Implications for Aging Research
Model architecture	Linear models with fixed feature relationships	Transformer-based model with contextual embeddings	Captures nonlinear gene–gene interactions relevant to aging
Data representation	Uses preselected or summarized features	Learns latent representations across tissues and conditions	Detects shared aging signatures
Interpretability	Statistically transparent but biologically limited: indicate correlation, not causation	More interpretable: perturbation analysis reveals gene influence on predicted age	Enables hypothesis generation about key regulatory genes
Generalizability	Dataset-specific; often tissue-limited	Pretrained on large multi-tissue data	Facilitates cross-tissue and cross-study generalization
Causal inference	Based on association only; weak causal inference	In silico perturbations directly test the model’s response to gene expression changes	Provides clues about potential pro- and anti-aging regulators
Computational requirements	Low; easily reproducible on standard hardware	Very high GPU and memory demand	Much higher costs but greater modeling capacity and discovery potential
Data requirements	Works with small, well-curated datasets	Benefits from large, heterogeneous single-cell datasets	Leverages large-scale data to uncover universal aging patterns
Experimental integration	Descriptive correlations without actionable targets	Provides ranked candidate genes for downstream perturbation studies	Bridges predictive modeling and experimental validation
Overall role	Descriptive and correlative	Predictive and hypothesis-generating	Advances from correlations to hypothesis generation

## Data Availability

Custom code created for this study is openly available at https://github.com/ifreyk/scgpt-age (accessed on 2 November 2025). The AgeAnno dataset analyzed in this study is openly available at https://relab.xidian.edu.cn/AgeAnno/#/ (accessed on 20 April 2024).

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
