# Peer review of "Discovering Candidate Anti-Aging Perturbations Using a Foundation Model for Gene Expression"

_ijms, 2025, doi:10.3390/ijms262411977_

Round 1
Reviewer 1 Report
Comments and Suggestions for Authors
This manuscript presents an exploratory study that applies the foundation model scGPT to age prediction using single-cell transcriptomic data, followed by in silico perturbation analyses to identify potential pro-aging and anti-aging factors. The topic is timely and innovative, and the methodological workflow is clearly outlined. However, several major issues remain, including insufficient confounding control, methodological limitations in the perturbation analysis, inadequate interpretation of model performance, and overstatement of biological conclusions.
Overall, substantial revision is required. I recommend Major Revision.
Major Points:
1.Although the authors describe the work as a “proof-of-concept”, the manuscript repeatedly employs terms such as “pro-aging” , “anti-aging” , “geroprotectors” and “geroaccelerators” which imply biological causality. The perturbation outcomes, however, only reflect internal model behavior and should not be interpreted mechanistically. My detailed comments are as follows:
- Explicitly differentiate between internal model representations and biological mechanism.
- Avoid terminology implying causality, or include strong disclaimers.
2.The AgeAnno dataset provides chronological age, not biological age.Furthermore, the manuscript does not examine age-tissue balance or potential confounding from tissue composition, which may lead the model to learn tissue identity rather than aging signatures.My detailed comments are as follows:
- Provide age distribution plots and age-by-tissue tables.
- Report the number of cells per tissue and age group.
- Perform tissue-wise or cell-type–wise evaluation.
- Discuss the conceptual limitations of chronological age as a label or incorporating biological age as another label.
3.Setting gene expression values to 0 or 10,000 raw counts likely pushes the model far outside the training distribution, raising concerns about prediction validity. No justification or expression distribution is provided.
4.The improvement from logistic regression (AUC 0.894) to scGPT (AUC 0.908) is minor. The authors should discuss whether the effect is meaningful, whether features are largely linear, and whether performance is inflated by tissue confounding. Additional evaluation plots are recommended.
- Key methodological details are missing, including data split strategy, seed control, frozen layers of scGPT, rationale for the 51 expression bins, and batch correction steps. These should be clarified for reproducibility.
Minor Points
- Related lines: 142-143
Explain why these 3,000 genes are used instead of the full gene set or alternative selection strategies.
- Related lines: 174-175
Clarify whether this cell number is sufficient for statistical robustness and whether results were stable across random subsampling.
- Related lines: 150-152、162
The formatting of text within parentheses (such as font style, weight, or size) is slightly inconsistent with the main text in several locations. Please ensure that all text inside parentheses matches the formatting of the surrounding body text to maintain uniformity and comply with journal formatting standards.
Reviewer 2 Report
Comments and Suggestions for Authors
In the proposed manuscript, the authors present a deep-learning framework that fine-tunes the scGPT transcriptomic model on multi-tissue single-cell aging data to predict age groups and identify candidate pro- and anti-aging genes through in-silico perturbations. The work is submitted as a brief report, is well written, and relatively easy to follow. The methodology is described in sufficient detail, and the code is available on GitHub. I have several questions and suggestions for the authors:
- To broaden the pool of potential readers, I suggest adding a few sentences that explain the rationale, approach, and main results in plain language. I do not mean an oversimplification, but just enough to enhance accessibility and readability.
-
Regarding the perturbation analysis, an illustrative example showing the effect of a known aging-related gene (e.g., Foxo3) on its pathway-related targets would be helpful to anchor the results in a familiar biological context.
- Can the authors comment on the AUC values for their approach compared with the LR model? Were they expecting a larger performance gap, and in their opinion, why does the trained model not outperform LR more prominently?
- Related to (3), what was the rationale for selecting 1,300 cells for evaluation?
- The first two sentences in Section 5 do not appear to fit well and may need revision.
- I suggest including an additional reference on single-cell aging clocks (PMID: 37524436).
Additional comment (no action needed):
I agree with the authors’ statement in the Discussion that additional experimental work would be required to more fully evaluate the performance and practical utility of their model. Ideally, such a complete study would be conducted and submitted as a unified manuscript; that would indeed be a very interesting read. However, I respect the authors' decision to present this work as a brief report.
Best regards
